# Fair Graph Message Passing

## Abstract

There has been significant progress in improving the performance of graph neural networks (GNNs) through enhancements in graph data, model architecture design, and training strategies. For fairness in graphs, recent studies achieve fair representations and predictions through either graph data pre-processing (e.g., node feature masking, and topology rewiring) or fair training strategies (e.g., regularization, adversarial debiasing, and fair contrastive learning). How to achieve fairness in graphs from the model architecture perspective is less explored. More importantly, GNNs exhibit worse fairness performance compared to multi-layer perception since their model architecture (i.e., neighbor aggregation) amplifies biases. To this end, we aim to achieve fairness via a new GNN architecture. We propose Fair Message Passing (FMP) designed within a unified optimization framework for GNNs. Notably, FMP *explicitly* rendering sensitive attribute usage in *forward propagation* for node classification task using cross-entropy loss without data pre-processing. In FMP, the aggregation is first adopted to utilize neighbors' information and then the bias mitigation step explicitly pushes demographic group node presentation centers together. In this way, FMP scheme can aggregate useful information from neighbors and mitigate bias to achieve better fairness and prediction tradeoff performance. Experiments on node classification tasks demonstrate that the proposed FMP outperforms several baselines in terms of fairness and accuracy on three real-world datasets. The code is available in https://anonymous.4open.science/r/FMP-AD84.

## 1 Introduction

Graph neural networks (GNNs) (Kipf & Welling, 2017; Veličković et al., 2018; Wu et al., 2019; Han et al., 2022a;b) are widely adopted in various domains, such as social media mining (Hamilton et al., 2017), knowledge graph (Hamaguchi et al., 2017) and recommender system (Ying et al., 2018), due to remarkable performance in learning representations. Graph learning, a topic with growing popularity, aims to learn node representation containing both topological and attribute information in a given graph. Despite the outstanding performance in various tasks, GNNs often inherit or even amplify societal bias from input graph data (Dai & Wang, 2021). The biased node representation largely limits the application of GNNs in many high-stake tasks, such as job hunting (Mehrabi et al., 2021) and crime ratio prediction (Suresh & Guttag, 2019). Hence, bias mitigation that facilitates the research on fair GNNs is in urgent need and we aim to achieve fair prediction for GNNs.

Data, model architecture, and training strategy are the most popular aspects to improve deep learning performance. For fairness in graphs, many existing works achieving fair prediction in graphs either rely on graph pre-processing (e.g., node feature masking(Köse & Shen, 2021), and topology rewiring (Dong et al., 2022)) or fair training strategies (e.g., regularization (Jiang et al., 2022), adversarial debiasing (Dai & Wang, 2021), or contrastive learning (Zhu et al., 2020; 2021b; Agarwal et al., 2021; Ling et al., 2023)). The GNNs architecture perspective to improve fairness in graphs is less explored. More importantly, GNNs are notorious in terms of fairness since GNN aggregation amplifies bias compared to multilayer perception (MLP) (Dai & Wang, 2021). From the GNNs architecture perspective, message passing is a critical component to improve fairness in graphs. Therefore, a natural question is raised:

*Can we achieve fairness via fair message passing using vanilla training loss [1] without graph pre-processing?*

In this work, we provide a positive answer by designing a fair message-passing scheme guided by a unified optimization framework [2] for GNNs. The key idea of achieving fair message passing is aggregation first and then conducting bias mitigation via explicitly chasing consistent demographic group representation centers. Specifically, we first formulate an optimization problem that integrates fairness and smoothness objectives for graph data. Then, we solve the formulated problem via Fenchel conjugate and gradient descent to generate fair and informative representations, where the property of softmax function is adopted to accelerate the gradient calculation over primal variables. We also interpret the optimization problem solver as two main steps (e.g., aggregation first and then debiasing). Finally, we integrate FMP in graph neural networks to achieve fair and accurate prediction for node classification tasks. We demonstrate the superiority of FMP by examining its effectiveness and efficiency on various real-world datasets.

In short, the contributions can be summarized as follows:

- We demonstrate proof-of-concept that a meticulously crafted GNN architecture can achieve fairness for graph data. Our work offers a fresh outlook in comparison to conventional approaches that focus on data pre-processing and fair training strategy design.

- We propose FMP to achieve fairness via explicitly incorporating sensitive attribute information in message passing, guided by a unified optimization framework. Additionally, we introduce an acceleration method based on softmax property to reduce gradient computational complexity.

- The effectiveness and efficiency of FMP are experimentally evaluated on three real-world datasets. The results show that compared to the state-of-the-art, our FMP exhibits a comparable or superior trade-off between prediction performance and fairness with negligibly computation overhead.

## 2 Preliminaries

### 2.1 Notations

We adopt bold upper-case letters to denote matrix such as $\mathbf{X}$, bold lower-case letters such as $\mathbf{x}$ to denote vectors, and calligraphic font such as $\mathcal{X}$ to denote sets. Given a matrix $\mathbf{X} \in \mathbb{R}^{n \times d}$, the $i$-th row and $j$-th column are denoted as $\mathbf{X}_i$ and $\mathbf{X}_{.,j}$, and the element in $i$-th row and $j$-th column is $\mathbf{X}_{i,j}$. We use the Frobenius norm, $l_1$ norm of matrix $\mathbf{X}$ as $||\mathbf{X}||_F = \sqrt{\sum_{i,j} \mathbf{X}_{i,j}^2}$ and $||\mathbf{X}||_1 = \sum_{ij} |\mathbf{X}_{ij}|$, respectively. Given two matrices $\mathbf{X}, \mathbf{Y} \in \mathbb{R}^{n \times d}$, the inner product is defined as $\langle \mathbf{X}, \mathbf{Y} \rangle = tr(\mathbf{X}^\top \mathbf{Y})$, where $tr(\cdot)$ is the trace of a square matrix. $SF(\mathbf{X})$ represents softmax function with a default normalized column dimension. Let $\mathcal{G} = \{\mathcal{V}, \mathcal{E}\}$ be a graph with the node set $\mathcal{V} = \{v_1, \cdots, v_n\}$ and the undirected edge set $\mathcal{E} = \{e_1, \cdots, e_m\}$, where $n, m$ represent the number of node and edge, respectively. The graph structure $\mathcal{G}$ can be represented as an adjacent matrix $\mathbf{A} \in \mathbb{R}^{n \times n}$, where $\mathbf{A}_{ij} = 1$ if existing edge between node $v_i$ and node $v_j$. $\mathcal{N}(i)$ denotes the neighbors of node $v_i$ and $\tilde{\mathcal{N}}(i) = \mathcal{N}(i) \cup \{v_i\}$ denotes the self-inclusive neighbors. Suppose that each node is associated with a $d$-dimensional feature vector and a (binary) sensitive attribute, the feature for all nodes and sensitive attribute is denoted as $\mathbf{X}_{ori} = \mathbb{R}^{n \times d}$ and $\mathbf{s} \in \{-1, 1\}^n$ [3]. Define the sensitive attribute incident vector as $\Delta_{\mathbf{s}} = \frac{\mathbb{1}_{>0}(\mathbf{s})}{||\mathbb{1}_{>0}(\mathbf{s})||_1} - \frac{\mathbb{1}_{>0}(-\mathbf{s})}{||\mathbb{1}_{>0}(-\mathbf{s})||_1}$ to normalize each sensitive attribute group, where $\mathbb{1}_{>0}(\mathbf{s})$ is an element-wise indicator function.

### 2.2 GNNs as Graph Signal Denoising

A GNN model is usually composed of several stacking GNN layers. Given a graph $\mathcal{G}$ with $N$ nodes, a GNN layer typically contains feature transformation $\mathbf{X}_{trans} = f_{trans}(\mathbf{X}_{ori})$ and aggregation $\mathbf{X}_{agg} = f_{agg}(\mathbf{X}_{trans}|\mathcal{G})$, where $\mathbf{X}_{ori} \in \mathbb{R}^{n \times d_{in}}$, $\mathbf{X}_{trans}, \mathbf{X}_{agg} \in \mathbb{R}^{n \times d_{out}}$ represent the input and output features. The

---

[1] The sensitive attributes are not adopted in vanilla training loss. We only consider node classification tasks and vanilla loss is cross-entropy loss in this paper.

[2] Many aggregations in popular GNNs can be interpreted as gradient descent step for specific optimization problem with specific step size and initialization (Ma et al., 2021b; Zhu et al., 2021b).

[3] The sensitive attribute $\mathbf{s}$ is not included in node features matrix $\mathbf{X}_{ori}$.

feature transformation operation transforms the node feature dimension, and *feature aggregation*, updates node features based on neighbors' features and graph topology. Recent works (Ma et al., 2021b; Zhu et al., 2021a) have established the connections between many feature aggregation operations $AGG(\cdot)$ in representative GNNs and a graph signal denoising problem with Laplacian regularization. Here, we introduce several popular GNN architectures, including GCN/SGC, GAT, and PPNP/APPNP, as examples to show the connection from the perspective of graph signal denoising.

**GCN/SGC.** Feature aggregation in Graph Convolutional Network (GCN) or Simplifying Graph Convolutional Network (SGC) is given by $\mathbf{X}_{agg} = \tilde{\mathbf{A}}\mathbf{X}_{trans}$, where $\tilde{\mathbf{A}} = \hat{\mathbf{D}}^{-\frac{1}{2}}\hat{\mathbf{A}}\tilde{\mathbf{D}}^{-\frac{1}{2}}$ is a normalized self-loop adjacency matrix $\hat{\mathbf{A}} = \mathbf{A} + \mathbf{I}$, and $\tilde{\mathbf{D}}$ is degree matrix of $\tilde{\mathbf{A}}$. Recent works (Ma et al., 2021b; Zhu et al., 2021a) provably demonstrate that such feature aggregation can be interpreted as one-step gradient descent to minimize $tr(\mathbf{F}^{\top}(\mathbf{I} - \tilde{\mathbf{A}})\mathbf{F})$ with initialization $\mathbf{F} = \mathbf{X}_{trans}$.

**GAT.** Feature aggregation in GAT applies the normalized attention coefficient to compute a linear combination of neighbor's features as $\mathbf{X}_{agg,i} = \sum_{j\in\mathcal{N}(i)} \alpha_{ij}\mathbf{X}_{trans,j}$, where $\alpha_{ij} = softmax_j(e_{ij})$, $e_{ij} = $ LeakyReLU$(\mathbf{X}_{trans,i}^{\top}\mathbf{w}_i + \mathbf{X}_{trans,j}^{\top}\mathbf{w}_j)$, and $\mathbf{w}_i$ and $\mathbf{w}_j$ are learnable column vectors. Prior study Ma et al. (2021b) demonstrates that one-step gradient descent with adaptive stepsize $\frac{1}{\sum_{j\in\tilde{\mathcal{N}}(i)}(c_i+c_j)}$ for the following objective problem:

$$\min_{\mathbf{F}} \sum_{i\in\mathcal{V}} ||\mathbf{F}_i - \mathbf{X}_{trans,i}||_F^2 + \frac{1}{2}\sum_{i\in\mathcal{V}} c_i \sum_{j\in\tilde{\mathcal{N}}(i)} ||\mathbf{F}_i - \mathbf{F}_j||_F^2.$$

is actually an attention-based feature aggregation, which is equivalent to GAT if $c_i + c_j$ is equivalent to $e_{ij}$, where $c_i$ is a node-dependent coefficient that measures the local smoothness.

**PPNP / APPNP.** Feature aggregation in PPNP and APPNP adopt the aggregation rules as $\mathbf{X}_{agg} = \alpha\left(\mathbf{I} - (1-\alpha)\tilde{\mathbf{A}}\right)^{-1}\mathbf{X}_{trans}$ and $\mathbf{X}_{agg}^{k+1} = (1-\alpha)\tilde{\mathbf{A}}\mathbf{X}_{agg}^k + \alpha\mathbf{X}_{trans}$. It is shown that they are equivalent to the exact solution and one gradient descent step with stepsize $\frac{\alpha}{2}$ to minimize the following objective problem:

$$\min_{\mathbf{F}} ||\mathbf{F} - \mathbf{X}_{trans}||_F^2 + (\frac{1}{\alpha} - 1)tr\left(\mathbf{F}^{\top}(\mathbf{I} - \tilde{\mathbf{A}})\mathbf{F}\right).$$

## 3 Fair Message Passing

In this section, we propose a new fair message-passing scheme to aggregate useful information from neighbors while debiasing representation bias. In this way, fair prediction can be achieved from a model backbone perspective. Specifically, we formulate fair message passing as an optimization problem to pursue *smoothness* and *fair* node representation simultaneously [4]. Together with an effective and efficient optimization algorithm, we derive the closed-form fair message passing. Finally, the proposed FMP is shown to be integrated into fair GNNs at three stages, including transformation, aggregation, and debiasing step, as shown in Figure 1. These three stages adopted node feature, graph topology, and sensitive attributes respectively.

### 3.1 The Optimization Framework

In previous work (Ma et al., 2021b), a general and universal framework is developed to understand aggregation operations in GNNs. Building on top of this framework, we formulate an optimization problem to achieve fair message passing operation (replace aggregation operations in GNNs). To achieve graph smoothness prior and fairness in the same process, a reasonable message passing should be a good solution for the following optimization problem:

$$\min_{\mathbf{F}} \underbrace{\frac{\lambda_s}{2}tr(\mathbf{F}^T\tilde{\mathbf{L}}\mathbf{F}) + \frac{1}{2}||\mathbf{F} - \mathbf{X}_{trans}||_F^2}_{h_s(\mathbf{F})} + \underbrace{\lambda_f||\mathbf{\Delta}_s SF(\mathbf{F})||_1}_{h_f\left(\mathbf{\Delta}_s SF(\mathbf{F})\right)}. \tag{1}$$

---

[4]Fair message passing is an alternative operation to replace GNNs aggregations.

where $\tilde{\mathbf{L}}$ represents normalized Laplacian matrix, $h_s(\cdot)$ and $h_f(\cdot)$ denotes the smoothness and fairness objectives [5], respectively, and $\mathbf{X}_{trans} \in \mathbf{R}^{n \times d_{out}}$ is the transformed $d_{out}$-dimensional node features and $\mathbf{F} \in \mathbf{R}^{n \times d_{out}}$ is the aggregated node features of the same matrix size. The first two terms preserve the similarity of connected node representation and thus enforce graph smoothness. The last term enforces fair node representation so that the average predicted probability between groups of different sensitive attributes can remain constant. The regularization coefficients $\lambda_s$ and $\lambda_f$ adaptively control the trade-off between graph smoothness and fairness.

**Smoothness Objective $h_s(\cdot)$.** The adjacent matrix in existing graph message passing schemes is normalized for improving numerical stability and achieving superior performance. Similarly, the graph smoothness term requires normalized Laplacian matrix, i.e., $\tilde{\mathbf{L}} = \mathbf{I} - \tilde{\mathbf{A}}$, $\tilde{\mathbf{A}} = \hat{\mathbf{D}}^{-\frac{1}{2}} \hat{\mathbf{A}} \hat{\mathbf{D}}^{-\frac{1}{2}}$, and $\hat{\mathbf{A}} = \mathbf{A} + \mathbf{I}$. From an edge-centric view, the smoothness objective enforces connected node representation to be similar since

$$tr(\mathbf{F}^T \tilde{\mathbf{L}} \mathbf{F}) = \sum_{(v_i,v_j) \in \mathcal{E}} ||\frac{\mathbf{F}_i}{\sqrt{d_i + 1}} - \frac{\mathbf{F}_j}{\sqrt{d_j + 1}}||_F^2, \tag{2}$$

where $d_i = \sum_k A_{ik}$ represents the degree of node $v_i$.

**Fairness Objective $h_f(\cdot)$.** The fairness objective measures the bias for node representation after aggregation. Recall sensitive attribute incident vector $\Delta_{\mathbf{s}}$ indicates the sensitive attribute group and group size via the sign and absolute value summation. Recall that the sensitive attribute incident vector as

$$\Delta_{\mathbf{s}} = \frac{\mathbb{1}_{>0}(\mathbf{s})}{||\mathbb{1}_{>0}(\mathbf{s})||_1} - \frac{\mathbb{1}_{>0}(-\mathbf{s})}{||\mathbb{1}_{>0}(-\mathbf{s})||_1}, \tag{3}$$

and $SF(\mathbf{F})$ represents the predicted probability for node classification task, where $SF(\mathbf{F})_{ij} = \hat{P}(y_i = j|\mathbf{X})$. Furthermore, we can show that our fairness objective is actually equivalent to demographic parity, i.e., $\left(\Delta_s SF(\mathbf{F}))\right)_j = \hat{P}(y_i = j|\mathbf{s}_i = 1, \mathbf{X}) - \hat{P}(y_i = j|\mathbf{s}_i = -1, \mathbf{X})$. Please see proof in Appendix B. In other words, our fairness objective, $l_1$ norm of $\Delta_s SF(\mathbf{F})$ characterizes the predicted probability difference between two groups with different sensitive attributes. Therefore, our proposed optimization framework can pursue graph smoothness and fairness simultaneously.

## 3.2 Optimization Problem Solver

For smoothness objective, many existing popular message passing schemes can be derived based on gradient descent with appropriate step size choice (Ma et al., 2021b; Zhu et al., 2021a). In this paper, we consider smoothness objective $h_s(\mathbf{F})$ and fairness objective $h_f(\Delta SF(\mathbf{F}))$ simultaneously for chasing fair and accurate prediction. However, directly solving the optimization problem (1) is much more challenging due to the nonsmoothness of the fairness objective, and the non-separability of smoothness objective $h_s(\mathbf{F})$ and fairness objective $h_f(\Delta SF(\mathbf{F}))$ due to incident vector $\Delta_s$.

### 3.2.1 Bi-level Optimization Problem Formulation.

In the literature, many optimization algorithms are developed for optimization problems with $l_1$ norm, such as Alternating Direction Method of Multipliers (ADMM) and Newton type algorithms (Ghadimi et al., 2014; Varma et al., 2019). However, these algorithms require non-trivial sub-problem solving for each iteration. Therefore, computation complexity is high and is infeasible to integrate deep learning models. Fortunately, Fenchel conjugate (a.k.a. convex conjugate) (Rockafellar, 2015) can transform the original problem as an equivalent saddle point problem using a primal-dual algorithm (Liu et al., 2021). In this way, the computation complexity can be reduced and compatible with back-propagation training. Similarly, to solve optimization problem 1 in a more effective and efficient manner, Fenchel conjugate (Rockafellar, 2015)

---

[5]Such smoothness objective is the most common-used one in existing methods (Ma et al., 2021b; Belkin & Niyogi, 2001; Kalofolias, 2016). The various other smoothness objectives could be considered to improve the performance of FMP and we leave it for future work.

is introduced to transform the original problem into a bi-level optimization problem. For the general convex function $h(\cdot)$, its conjugate function is defined as $h^*(\mathbf{U}) \triangleq \sup_{\mathbf{X}} \langle \mathbf{U}, \mathbf{X} \rangle - h(\mathbf{X})$. Based on Fenchel conjugate, the fairness objective can be transformed as variational representation $h_f(\mathbf{p}) = \sup_{\mathbf{u}} \langle \mathbf{p}, \mathbf{u} \rangle - h_f^*(\mathbf{u})$, where $\mathbf{p} = \mathbf{\Delta}_s SF(\mathbf{F}) \in \mathbb{R}^{1 \times d_{out}}$ is a predicted probability vector for classification. Furthermore, the original optimization problem is equivalent to

$$\min_{\mathbf{F}} \max_{\mathbf{u}} h_s(\mathbf{F}) + \langle \mathbf{p}, \mathbf{u} \rangle - h_f^*(\mathbf{u}) \tag{4}$$

where $\mathbf{u} \in \mathbb{R}^{1 \times d_{out}}$ and $h_f^*(\cdot)$ is the conjugate function of fairness objective $h_f(\cdot)$.

### 3.2.2 Problem Solution

Motivated by Proximal Alternating Predictor-Corrector (PAPC) (Loris & Verhoeven, 2011; Chen et al., 2013), the min-max optimization problem (4) can be solved by the following fixed-point equations with per iteration low computation complexity and convergence guarantee

$$\begin{cases} \mathbf{F} = \mathbf{F} - \nabla h_s(\mathbf{F}) - \frac{\partial \langle \mathbf{p}, \mathbf{u} \rangle}{\partial \mathbf{F}}, \\ \mathbf{u} = \operatorname{prox}_{h_f^*}\big(\mathbf{u} + \mathbf{\Delta_s} SF(\mathbf{F})\big). \end{cases} \tag{5}$$

where $\operatorname{prox}_{h_f^*}(\mathbf{u}) = \arg \min_{\mathbf{y}} ||\mathbf{y} - \mathbf{u}||_F^2 + h_f^*(\mathbf{y})$. Fortunately, the proximal operators can be obtained with a close form, which makes deep learning model integration feasible. Specifically we provide the close form of the proximal operators in the following proposition:

**Proposition 3.1** (Proximal Operators). *The proximal operators $prox_{\beta h_f^*}(\mathbf{u})$ satisfies*

$$prox_{\beta h_f^*}(\mathbf{u})_j = sign(\mathbf{u})_j \min\big(|\mathbf{u}_j|, \lambda_f\big), \tag{6}$$

*where $sign(\cdot)$ and $\lambda_f$ are element-wise sign function and hyperparameter for fairness objective. In other words, such a proximal operator is an element-wise projection into $l_\infty$ ball with radius $\lambda_f$.*

Similar to "predictor-corrector" algorithm (Loris & Verhoeven, 2011), we adopt an iterative algorithm to find the saddle point for the min-max optimization problem. Specifically, starting from $(\mathbf{F}^k, \mathbf{u}^k)$, we adopt a gradient descent step on the primal variable $\mathbf{F}$ to arrive $(\bar{\mathbf{F}}^{k+1}, \mathbf{u}^k)$ and then followed by a proximal ascent step in the dual variable $\mathbf{u}$. Finally, a gradient descent step on a primal variable in point $(\bar{\mathbf{F}}^{k+1}, \mathbf{u}^k)$ to arrive at $(\mathbf{F}^{k+1}, \mathbf{u}^k)$. In short, the iteration can be summarized as

$$\begin{cases} \bar{\mathbf{F}}^{k+1} = \mathbf{F}^k - \gamma \nabla h_s(\mathbf{F}^k) - \gamma \frac{\partial \langle \mathbf{p}, \mathbf{u}^k \rangle}{\partial \mathbf{F}}\Big|_{\mathbf{F}^k}, \\ \mathbf{u}^{k+1} = \operatorname{prox}_{\beta h_f^*}\big(\mathbf{u}^k + \beta \mathbf{\Delta_s} SF(\bar{\mathbf{F}}^{k+1})\big), \\ \bar{\mathbf{F}}^{k+1} = \mathbf{F}^k - \gamma \nabla h_s(\mathbf{F}^k) - \gamma \frac{\partial \langle \mathbf{p}, \mathbf{u}^{k+1} \rangle}{\partial \mathbf{F}}\Big|_{\mathbf{F}^k}. \end{cases} \tag{7}$$

where $\gamma$ and $\beta$ are the step size for primal and dual variables. Note that the close-form for $\frac{\partial \langle \mathbf{p}, \mathbf{u} \rangle}{\partial \mathbf{F}} \in \mathbb{R}^{n \times d_{out}}$ and $\operatorname{prox}_{\beta h_f^*}(\cdot)$ are still not clear, we will provide the solution one by one.

**FMP Scheme.** Similar to works (Ma et al., 2021b; Liu et al., 2021), choosing $\gamma = \frac{1}{1+\lambda_s}$ and $\beta = \frac{1}{2\gamma}$, we have

$$\begin{aligned} \mathbf{F}^k - \gamma \nabla h_s(\mathbf{F}^k) &= \Big((1-\gamma)\mathbf{I} - \gamma \lambda_s \tilde{\mathbf{L}}\Big) \mathbf{F}^k + \gamma \mathbf{X}_{trans} \\ &= \gamma \mathbf{X}_{trans} + (1-\gamma)\tilde{\mathbf{A}} \mathbf{F}^k, \end{aligned} \tag{8}$$

Therefore, we can summarize the proposed FMP as two phases, including propagation with skip connection (Step ❶) and bias mitigation (Steps ❷-❺). For bias mitigation, Step ❷ updates the aggregated node features for fairness objective; Steps ❸ and ❹ aim to learn and "reshape" perturbation vector in probability space,

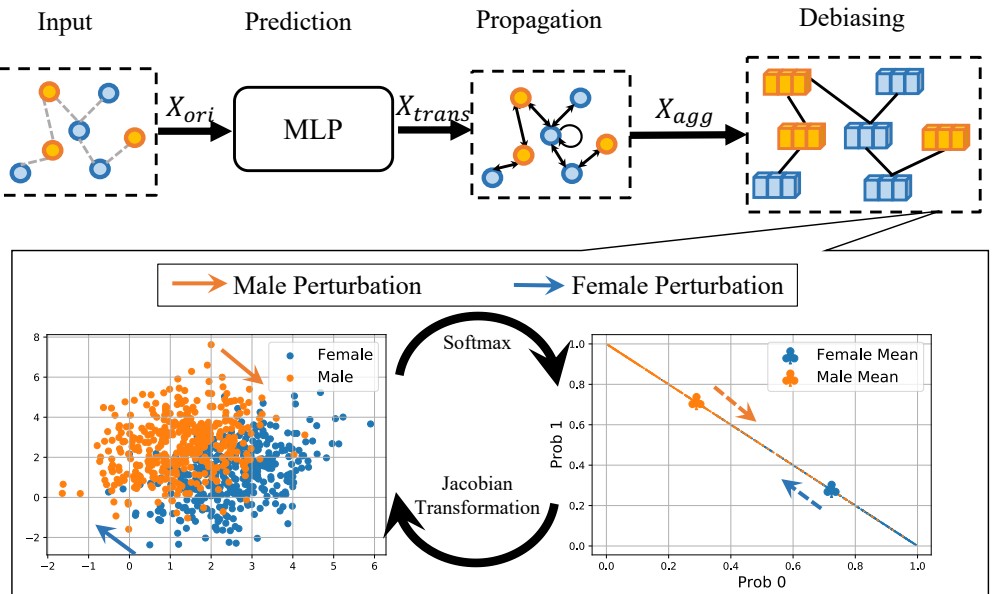

Figure 1: The model pipeline consists of three steps: MLP (feature transformation), propagation with skip connection, and debiasing via low-rank perturbation in probability space.

respectively. Step ❺ explicitly mitigates the bias of node features based on gradient descent on the primal variable. The mathematical formulation is given as follows:

$$\begin{cases} \mathbf{X}_{agg}^{k+1} = \gamma \mathbf{X}_{trans} + (1-\gamma)\tilde{\mathbf{A}}\mathbf{F}^k, & \text{Step ❶} \\ \bar{\mathbf{F}}^{k+1} = \mathbf{X}_{agg}^{k+1} - \gamma \frac{\partial\langle\mathbf{p},\mathbf{u}^k\rangle}{\partial\mathbf{F}}\Big|_{\mathbf{F}^k}, & \text{Step ❷} \\ \bar{\mathbf{u}}^{k+1} = \mathbf{u}^k + \beta \boldsymbol{\Delta}_\mathbf{s} SF(\bar{\mathbf{F}}^{k+1}), & \text{Step ❸} \\ \mathbf{u}^{k+1} = \min\left(|\bar{\mathbf{u}}^{k+1}|, \lambda_f\right) \cdot sign(\bar{\mathbf{u}}^{k+1}), & \text{Step ❹} \\ \mathbf{F}^{k+1} = \mathbf{X}_{agg}^{k+1} - \gamma \frac{\partial\langle\mathbf{p},\mathbf{u}^{k+1}\rangle}{\partial\mathbf{F}}\Big|_{\mathbf{F}^k}. & \text{Step ❺} \end{cases}$$

where $\mathbf{X}_{agg}^{k+1}$ represents the node features with normal aggregation and skip connection with the transformed input $\mathbf{X}_{trans}$.

### 3.2.3 Gradient Computation Acceleration

The softmax property is also adopted to accelerate the gradient computation. Note that $\mathbf{p} = \boldsymbol{\Delta}_s SF(\mathbf{F})$ and $SF(\cdot)$ represents softmax over column dimension, directly computing the gradient $\frac{\partial\langle\mathbf{p},\mathbf{u}\rangle}{\partial\mathbf{F}}$ based on chain rule involves the three-dimensional tensor $\frac{\partial\mathbf{p}}{\partial\mathbf{F}}$ with gigantic computation complexity. Instead, we simplify the gradient computation based on the property of softmax function in the following theorem.

**Theorem 3.2** (Gradient Computation). *The gradient over primal variable* $\frac{\partial\langle\mathbf{p},\mathbf{u}\rangle}{\partial\mathbf{F}}$ *satisfies*

$$\frac{\partial\langle\mathbf{p},\mathbf{u}\rangle}{\partial\mathbf{F}} = \mathbf{U}_s \odot SF(\mathbf{F}) - Sum_1(\mathbf{U}_s \odot SF(\mathbf{F}))SF(\mathbf{F}). \tag{9}$$

*where* $\mathbf{U}_s \overset{\triangle}{=} \Delta_s^\top \mathbf{u}$, $\odot$ *represents the element-wise product and* $Sum_1(\cdot)$ *represents the summation over column dimension with preserved matrix shape.*

## 4 Discussion on FMP

In this section, we provide the interpretation and analyze the *efficiency*, and *white-box usage for sensitive attribute* of the proposed FMP scheme. Furthermore, we also discuss how FMP identifies the influence of sensitive attributes from model forward propagation.

**FMP Interpretation**  Note that the gradient of fairness objective over node features $\mathbf{F}$ satisfies $\frac{\partial \langle \mathbf{p}, \mathbf{u} \rangle}{\partial \mathbf{F}} = \frac{\partial \langle \mathbf{p}, \mathbf{u} \rangle}{\partial SF(\mathbf{F})} \frac{\partial SF(\mathbf{F})}{\partial \mathbf{F}}$ and $\frac{\partial \langle \mathbf{p}, \mathbf{u} \rangle}{\partial SF(\mathbf{F})} = \Delta_s^\top \mathbf{u}$, such gradient calculation can be interpreted as three steps: Softmax transformation, perturbation in probability space, and debiasing in representation space. Specifically, we first map the node representation into probability space via softmax transformation. Subsequently, we calculate the gradient of fairness objective in probability space. It is seen that the perturbation $\Delta_s^\top \mathbf{u}$ actually poses *low-rank* debiasing in probability space, where the nodes with different sensitive attributes embrace opposite perturbations. In other words, *the dual variable $\mathbf{u}$ represents the perturbation direction in probability space*. Finally, the perturbation in probability space will be transformed into representation space via Jacobian transformation $\frac{\partial SF(\mathbf{F})}{\partial \mathbf{F}}$.

**Efficiency.**  FMP is an efficient message-passing scheme. The computation complexity for the aggregation (sparse matrix multiplications) is $O(md_{out})$, where $m$ is the number of edges in the graph. For FMP, the extra computation mainly focuses on the perturbation calculation, as shown in Theorem 3.2, with the computation complexity $O(nd_{out})$. The extra computation complexity is negligible in that the number of nodes $n$ is far less than the number of edges $m$ in the real-world graph. Additionally, if directly adopting backward propagation to calculate the gradient, we have to calculate the three-dimensional tensor $\frac{\partial \mathbf{p}}{\partial \mathbf{F}}$ with computation complexity $O(n^2 d_{out})$. In other words, thanks to the softmax property, we achieve an efficient fair message-passing scheme.

**White-box Usage for Sensitive Attribute.**  The proposed FMP explicitly achieves graph smoothness and fairness objectives via alternative gradient descent. In other words, the usage of sensitive attributes in propagation to mitigate bias is in a white-box manner. Note that such white-box usage of sensitive attributes is a promising property to understand how sensitive attribute usage forces fairness, which is not achieved by previous fairness methods in GNNs. For example, fair training loss utilizes sensitive attributes to regularize the behavior of model prediction and obtain fairer model parameters via rectifying gradients w.r.t. model parameters. In other words, the sensitive attribute information is implicitly encoded in the well-trained model parameters, which makes it hard to understand how sensitive attribute usage helps fair prediction. Pre-processing fairness methods adopt sensitive attributes to revise data (e.g., node masking and topology rewiring) either in a learnable way or via pre-defined several operations (e.g., node masking and edge deletions). Similarly, the sensitive attribute information is implicitly encoded in the processed data. The understanding of fairness prediction achievement is infeasible. Our FMP can provide a white-box usage for sensitive attributes since we can directly identify that the usage of sensitive attributes is to force the demographic group node representation centers together during forward propagation.

To facilitate the understanding of the influence of sensitive attributes, we measure the influence of sensitive attributes as the difference of final prediction between the well-trained fair model using sensitive attributes and vanilla models without sensitive attribute usage. The sensitive attribute has a critical influence to achieve fair prediction and the prediction is highly different for the vanilla model (trained with vanilla loss and no data preprocessing) and the fair model (trained with fair methods). We visualize the logit layer node representation for different methods in Appendix H.3.

The proposed FMP explicitly uses the sensitive attribute information in Steps ❷-❺ during forward propagation. In other words, if we aim to identify the influence of sensitive attributes for FMP, it is sufficient to check the difference between the input and output for the debiasing step since it is disentangled with feature transformation and aggregation. It is worth mentioning that the required information for identifying the influence of sensitive attributes is naturally from the forward propagation. However, for the fair model from existing works (e.g, adding regularization and adversarial debiasing), note that the sensitive attribute information is implicitly encoded in the well-trained model weight, the sensitive attribute perturbation inevitably leads to the variability of well-trained model weight. Therefore, it is required to retrain the model for probing the influence of sensitive attribute perturbation. The key drawback of these methods is due to encoding the sensitive attributes information into well-trained model weights. From the auditors' perspective, it is quite hard to identify the influence of sensitive attributes only given a well-trained fair model. Instead, our designed FMP explicitly adopts the sensitive attribute information in the forward propagation process, which naturally avoid the dilemma that sensitive attributes are encoded into well-trained model weight. In a nutshell, FMP encompasses higher transparency since (1) the sensitive attribute is explicitly adopted in

Table 1: Comparative Results with Baselines on Node Classification.

| Models | Pokec-z | | | Pokec-n | | | NBA | | |
|---|---|---|---|---|---|---|---|---|---|
| | Acc (%) ↑ | $\Delta_{DP}$ (%) ↓ | $\Delta_{EO}$ (%) ↓ | Acc (%) ↑ | $\Delta_{DP}$ (%) ↓ | $\Delta_{EO}$ (%) ↓ | Acc (%) ↑ | $\Delta_{DP}$ (%) ↓ | $\Delta_{EO}$ (%) ↓ |
| MLP | 70.48 ± 0.77 | 1.61 ± 1.29 | 2.22 ± 1.01 | 72.48 ± 0.26 | 1.53 ± 0.89 | 3.39 ± 2.37 | 65.56 ± 1.62 | 22.37 ± 1.87 | 18.00 ± 3.52 |
| GAT | 69.76 ± 1.30 | 2.39 ± 0.62 | 2.91 ± 0.97 | 71.00 ± 0.48 | 3.71 ± 2.15 | 7.50 ± 2.88 | 57.78 ± 10.65 | 20.12 ± 16.18 | 13.00 ± 13.37 |
| GCN | **71.78** ± 0.37 | 3.25 ± 2.35 | 2.36 ± 2.09 | **73.09** ± 0.28 | 3.48 ± 0.47 | 5.16 ± 1.38 | 61.90 ± 1.00 | 23.70 ± 2.74 | 17.50 ± 2.63 |
| SGC | 71.24 ± 0.46 | 4.81 ± 0.30 | 4.79 ± 2.27 | 71.46 ± 0.41 | 2.22 ± 0.29 | 3.85 ± 1.63 | 63.17 ± 0.63 | 22.56 ± 3.94 | 14.33± 2.16 |
| APPNP | 66.91 ± 1.46 | 3.90 ± 0.69 | 5.71 ± 1.29 | 69.80 ± 0.89 | 1.98 ± 1.30 | 4.01 ± 2.36 | 63.80 ± 1.19 | 26.51 ± 3.33 | 20.00 ± 4.56 |
| JKNet | 66.89 ± 3.79 | 1.28 ±0.96 | 1.79 ± 0.82 | 63.59 ± 6.36 | 1.91 ± 2.14 | **0.70** ± 0.92 | 67.94 ± 2.73 | 27.80 ± 8.41 | 20.33 ± 7.52 |
| ML1 | 70.42 ± 0.40 | 2.35 ± 0.83 | 2.00 ± 0.50 | 72.36 ± 0.26 | 1.47 ± 1.12 | 3.03 ± 1.77 | 72.70 ± 1.19 | 26.46 ± 4.93 | 25.50 ± 8.38 |
| FMP | 70.50 ± 0.50 | **0.81** ± 0.40 | **1.73** ± 1.03 | 72.16 ± 0.33 | **0.66** ± 0.40 | 1.47 ± 0.87 | **73.33** ± 1.85 | **18.92** ± 2.28 | **13.33** ± 5.89 |

forward propagation; (2) It is not necessary to retrain the model for probing the influence of the sensitive attribute.

## 5 Experiments

In this section, we conduct experiments to validate the effectiveness and efficiency of the proposed FMP. We firstly validate that graph data with large sensitive homophily enhances bias in GNNs via synthetic experiments. Moreover, for experiments on real-world datasets, we introduce the experimental settings and then evaluate our proposed FMP compared with several baselines in terms of prediction performance and fairness metrics.

### 5.1 Experimental Settings

**Datasets.** We conduct experiments on real-world datasets Pokec-z, Pokec-n, and NBA (Dai & Wang, 2021). Pokec-z and Pokec-n are sampled, based on province information, from a larger Facebook-like social network Pokec (Takac & Zabovsky, 2012) in Slovakia, where region information is treated as the sensitive attribute and the predicted label is the working field of the users. NBA dataset is extended from a Kaggle dataset [6] consisting of around 400 NBA basketball players. The information of players includes age, nationality, and salary in the 2016-2017 season. The players' link relationships are from Twitter with the official crawling API. The binary nationality (U.S. and overseas player) is adopted as the sensitive attribute and the prediction label is whether the salary is higher than the median.

**Evaluation Metrics.** We adopt accuracy to evaluate the performance of node classification tasks. As for fairness metrics, we adopt two quantitative group fairness metrics to measure the prediction bias. According to works (Louizos et al., 2015; beu), we adopt *demographic parity* $\Delta_{DP} = |\mathbb{P}(\hat{y} = 1|s = -1) - \mathbb{P}(\hat{y} = 1|s = 1)|$ and *equal opportunity* $\Delta_{EO} = |\mathbb{P}(\hat{y} = 1|s = -1, y = 1) - \mathbb{P}(\hat{y} = 1|s = 1, y = 1)|$, where $y$ and $\hat{y}$ represent the ground-truth label and predicted label, respectively.

**Baselines.** We compare our proposed FMP with representative GNNs, such as GCN (Kipf & Welling, 2017), GAT (Veličković et al., 2018), SGC (Wu et al., 2019), and APPNP (Klicpera et al., 2019), JKNet (Xu et al., 2018), and MLP. We also compared with method "ML1" directly using the gradient of Eq. (1) during model forward propagation. For all models, we train 2 layers of neural networks with 64 hidden units for 300 epochs. Additionally, We also compare adversarial debiasing and adding demographic regularization methods to show the effectiveness of the proposed method [7].

**Implementation Details.** We run the experiments 5 times and report the average performance for each method. We adopt Adam optimizer with 0.001 learning rate and $10^{-5}$ weight decay for all models. For adversarial debiasing, we adopt the train classifier and adversary with 70 and 30 epochs, respectively. The

---

[6]https://www.kaggle.com/noahgift/social-power-nba
[7]Please see the comparison with Fair Mixup (Chuang & Mroueh, 2021) in Appendix H.2

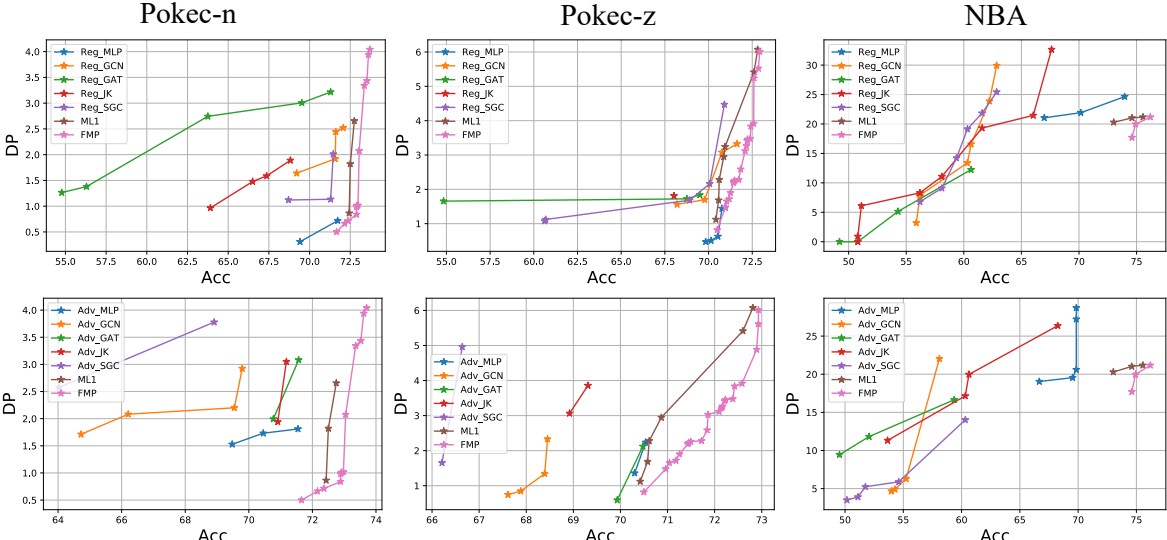

Figure 2: DP and Acc trade-off performance on three real-world datasets compared with adding regularization (Top) and adversarial debiasing (Bottom). The trade-off curve close to the right bottom corner means better trade-off performance. The units for x- and y-axis are percentages (%).

hyperparameter for adversary loss is tuned in $\{0.0, 1.0, 2.0, 5.0, 8.0, 10.0, 20.0, 30.0\}$. For adding regularization, we adopt the hyperparameter set $\{0.0, 1.0, 2.0, 5.0, 8.0, 10.0, 20.0, 50.0, 80.0, 100.0\}$.

## 5.2 Experimental Results

**Comparison with Existing GNNs.** The accuracy, demographic parity, and equal opportunity metrics of proposed FMP for Pokec-z, Pokec-n, NBA datasets are shown in Table 1 compared with MLP, GAT, GCN, SGC, and APPNP. The detailed statistical information for these three datasets is shown in Table 3. From these results, we can obtain the following observations:

- Many existing GNNs underperform MLP model on all three datasets in terms of fairness metric. For instance, the demographic parity of MLP is lower than GAT, GCN, SGC and APPNP by 32.64%, 50.46%, 66.53% and 58.72% on Pokec-z dataset. The higher prediction bias comes from the aggregation within the same sensitive attribute nodes and topology bias in graph data.

- Our proposed FMP consistently achieves the lowest prediction bias in terms of demographic parity and equal opportunity on all datasets. Specifically, FMP reduces demographic parity by 49.69%, 56.86%, and 5.97% compared with the lowest bias among all baselines in Pokec-z, Pokec-n, and NBA datasets. Meanwhile, our proposed FMP achieves the best accuracy in NBA dataset, and comparable accuracy in Pokec-z and Pokec-n datasets. In a nutshell, the proposed FMP can effectively mitigate prediction bias while preserving the prediction performance.

**Comparison with Adversarial Debiasing and Regularization.** To validate the effectiveness of the proposed FMP, we also show the prediction performance and fairness metric trade-off compared with fairness-boosting methods, including adversarial debiasing (Fisher et al., 2020) and adding regularization (Chuang & Mroueh, 2020). Similar to (lou), the output of GNNs is the input of the adversary and the goal of the adversary is to predict the node sensitive attribute. We also adopt several backbones for these two methods, including MLP, GCN, GAT, and SGC. We randomly split 50%/25%/25% for training, validation, and test dataset. Figure 2 shows the Pareto optimality curve for all methods, where the right-bottom corner point represents the ideal performance (highest accuracy and lowest prediction bias). From the results, we list the following observations as follows:

- Our proposed FMP can achieve better DP-Acc trade-off compared with adversarial debiasing and adding regularization for many GNNs and MLP. Such observation validates the effectiveness of the key idea in FMP: aggregation first and then debiasing. Additionally, FMP can reduce demographic parity with negligible performance cost due to transparent and efficient debiasing.

- Message passing in GNNs does matter. For adding regularization or adversarial debiasing, different GNNs embrace huge distinctions, which implies that an appropriate message passing manner potentially leads to better trade-off performance. Additionally, many GNNs underperforms MLP in low-label homophily coefficient dataset, such as NBA. The rationale is that aggregation may not always bring benefit in terms of accuracy when the neighbors have low probability with the same label.

## 6 Related Works

**Graph Neural Networks.** GNNs generalizing neural networks for graph data have already shown great success in various real-world applications. There are two streams in GNNs model design, i.e., spectral-based and spatial-based. Spectral-based GNNs provide graph convolution definition based on graph theory, which is utilized in GNN layers together with feature transformation (Bruna et al., 2013; Defferrard et al., 2016; Henaff et al., 2015). Graph convolutional networks (GCN) (Kipf & Welling, 2017) simplify spectral-based GNN model into spatial aggregation scheme. Since then, many spatial-based GNNs variant is developed to update node representation via aggregating its neighbors' information, including graph attention network (GAT) (Veličković et al., 2018), GraphSAGE (Hamilton et al., 2017), SGC (Wu et al., 2019), APPNP (Klicpera et al., 2019), et al (Gao et al., 2018; Monti et al., 2017). Graph signal denoising is another perspective to understand GNNs. Recently, there are several works show that GCN is equivalent to the first-order approximation for graph denoising with Laplacian regularization (Henaff et al., 2015; Zhao & Akoglu, 2019). The unified optimization framework is provided to unify many existing message passing schemes (Ma et al., 2021b; Zhu et al., 2021a).

**Fairness-aware Learning on Graphs.** Many works have been developed to achieve fairness in machine learning community (Jiang et al., 2022; Han et al., 2023; Jiang et al., 2023; Chuang & Mroueh, 2020; Zhang et al., 2018; Du et al., 2021; Yurochkin & Sun, 2020; Creager et al., 2019; Feldman et al., 2015). A pilot study on fair node representation learning is developed based on random walk (Rahman et al., 2019). Additionally, adversarial debiasing is adopted to learn fair prediction or node representation so that the well-trained adversary can not predict the sensitive attribute based on node representation or prediction (Dai & Wang, 2021; Bose & Hamilton, 2019; Fisher et al., 2020). A Bayesian approach is developed to learn fair node representation via encoding sensitive information in the prior distribution in (Buyl & De Bie, 2020). Work (Ma et al., 2021a) develops a PAC-Bayesian analysis to connect subgroup generalization with accuracy parity. (Laclau et al., 2021; Li et al., 2021) aims to mitigate prediction bias for link prediction. Fairness-aware graph contrastive learning is proposed in (Agarwal et al., 2021; Köse & Shen, 2021; Ling et al., 2023). Graph data preprocessing, such as node feature masking and graph topology rewire, are also developed in (Laclau et al., 2021; Li et al., 2021; Dong et al., 2021; 2023) for node classification and link prediction tasks. However, the aforementioned works ignore the requirement of transparency in fairness. In this work, we develop an efficient and transparent fair message passing scheme explicitly rendering sensitive attribute usage.

## 7 Conclusion

In this work, we achieve fairness in graphs from the model architecture perspective. We design a fair message-passing scheme to achieve fair prediction for node classification using vanilla training loss without data pre-processing. Specifically, motivated by the unified optimization framework for GNNs, FMP is designed as aggregation first and then bias mitigation to explicitly chase smoothness and fairness objectives. We also provide a comprehensive discussion of FMP from model architecture interpretation, efficiency, and the white-box usage of sensitive attributes aspects. Experimental results on real-world datasets demonstrate the effectiveness of FMP compared with several baselines in node classification tasks.

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

## A  Notations

Table 2: Table of Notations

| Notations | Description |
|---|---|
| $|\mathcal{E}|$ | The number of edges |
| $n$ | The number of nodes |
| $d$ | The number of node feature dimensions |
| $d_{out}$ | The number of node classes |
| $\Delta_{\mathbf{s}} \in \mathbb{R}^{1 \times n}$ | The sensitive attribute incident vector |
| $\epsilon_{label}$ | Label homophily coefficient |
| $\epsilon_{sens}$ | Sensitive homophily coefficient |
| $\mathbf{X}_{ori} \in \mathbb{R}^{n \times d}$ | The input node attributes matrix |
| $\mathbf{A} \in \mathbb{R}^{n \times n}$ | The adjacency matrix |
| $\hat{\mathbf{A}} \in \mathbb{R}^{n \times n}$ | The adjacency matrix with self-loop |
| $\tilde{\mathbf{A}} \in \mathbb{R}^{n \times n}$ | The normalized adjacency matrix with self-loop |
| $\mathbf{L} \in \mathbb{R}^{n \times n}$ | The Laplacian matrix |
| $\mathbf{X}_{trans} \in \mathbb{R}^{n \times d_{out}}$ | The output node features for feature transformation |
| $\mathbf{F}_{agg} \in \mathbb{R}^{n \times d_{out}}$ | The aggregated node features after propagation |
| $\mathbf{F} \in \mathbb{R}^{n \times d_{out}}$ | The learned node features considering graph smoothness and fairness |
| $\mathbf{u} \in \mathbb{R}^{1 \times d_{out}}$ | The permutation direction in feature representation space |
| $h^*(\cdot)$ | Fenchel conjugate function of $h(\cdot)$ |
| $||\mathbf{X}||_F, ||\mathbf{X}||_1$ | The Frobenius norm and $l_1$ norm of matrix $\mathbf{X}$ |
| $\lambda_f, \lambda_s$ | Hyperparameter for fairness and graph smoothness objectives |

## B  Proof on Fairness Objective

The fairness objective can be shown as the average prediction probability difference as follows:

$$
\begin{aligned}
\left( \Delta_s SF(\mathbf{F}) \right)_j &= \left[ \frac{\mathbb{1}_{>0}(\mathbf{s})}{||\mathbb{1}_{>0}(\mathbf{s})||_1} - \frac{\mathbb{1}_{>0}(-\mathbf{s})}{||\mathbb{1}_{>0}(-\mathbf{s})||_1} \right] \left( SF(\mathbf{F}) \right)_{:,j} \\
&= \frac{\sum_{\mathbf{s}_i=1} \hat{P}(y_i = j|\mathbf{X})}{||\mathbb{1}_{>0}(\mathbf{s})||_1} - \frac{\sum_{\mathbf{s}_i=-1} \hat{P}(y_i = j|\mathbf{X})}{||\mathbb{1}_{>0}(-\mathbf{s})||_1} \\
&= \hat{P}(y_i = j|\mathbf{s}_i = 1, \mathbf{X}) - \hat{P}(y_i = j|\mathbf{s}_i = -1, \mathbf{X}).
\end{aligned}
$$

## C  Proof of Theorem 3.2

Before providing in-depth analysis on the gradient computation, we first introduce the softmax function derivative property in the following lemma:

**Lemma C.1.** *For the softmax function with $N$-dimensional vector input $\mathbf{y} = SF(\mathbf{x}) : \mathbb{R}^{1 \times N} \longrightarrow \mathbb{R}^{1 \times N}$, where $y_j = \frac{e^{\mathbf{x}_j}}{\sum_{k=1}^{N} e^{\mathbf{x}_k}}$ for $\forall j \in \{1, 2, \cdots, N\}$, the derivative $N \times N$ Jocobian matrix is defined by $[\frac{\partial \mathbf{y}}{\partial \mathbf{x}}]_{ij} = \frac{\partial y_i}{\partial \mathbf{x}_j}$. Additionally, Jocobian matrix satisfies $\frac{\partial \mathbf{y}}{\partial \mathbf{x}} = \text{diag}(\mathbf{y}) - \mathbf{y}^\top \mathbf{y}$, where $\mathbf{I}_N$ represents $N \times N$ identity matrix and $\top$ denotes the transpose operation for vector or matrix.*

*Proof.* Considering the gradient $\frac{\partial \mathbf{y}_i}{\partial \mathbf{x}_j}$ for arbitrary $i = j$, according to quotient and chain rule of derivatives, we have

$$
\frac{\partial \mathbf{y}_i}{\partial \mathbf{x}_j} = \frac{e^{\mathbf{x}_i} \sum_{k=1}^{N} e^{\mathbf{x}_k} - e^{\mathbf{x}_i + \mathbf{x}_j}}{\left( \sum_{k=1}^{N} e^{\mathbf{x}_k} \right)^2} = \frac{e^{\mathbf{x}_i}}{\sum_{k=1}^{N} e^{\mathbf{x}_k}} \cdot \frac{\sum_{k=1}^{N} e^{\mathbf{x}_k} - e^{\mathbf{x}_i}}{\sum_{k=1}^{N} e^{\mathbf{x}_k}} = \mathbf{y}_i(1 - \mathbf{y}_j), \tag{10}
$$

Similarly, for arbitrary $i \neq j$, the gradient is given by

$$\frac{\partial \mathbf{y}_i}{\partial \mathbf{x}_j} = \frac{e^{\mathbf{x}_i}}{\sum_{k=1}^{N} e^{\mathbf{x}_k}} \cdot \frac{-e^{\mathbf{x}_i}}{\sum_{k=1}^{N} e^{\mathbf{x}_k}} = -\mathbf{y}_i \mathbf{y}_j. \tag{11}$$

Combining these two cases, it is easy to verify the Jacobian matrix satisfies $\frac{\partial \mathbf{y}}{\partial \mathbf{x}} = \mathrm{diag}(\mathbf{y}) - \mathbf{y}^\top \mathbf{y}$. □

Arming with the derivative property of softmax function, we further investigate the gradient $\frac{\partial \langle \mathbf{p}, \mathbf{u} \rangle}{\partial \mathbf{F}}$, where $\mathbf{p} = \boldsymbol{\Delta}_s SF(\mathbf{F}) \in \mathbb{R}^{1 \times d_{out}}$ and $SF(\cdot)$ and $\mathbf{u} \in \mathbb{R}^{1 \times d_{out}}$ is independent with $\mathbf{F} \in \mathbb{R}^{n \times d_{out}}$.

Considering softmax function $SF(\mathbf{x}) \in \mathbb{R}^{n \times d}$ is row-wise adopted in node representation matrix, the gradient satisfies $\frac{\partial SF(\mathbf{F})_i}{\partial \mathbf{F}_j} = \mathbf{0}_{d_{out} \times d_{out}}$ for $i \neq j$. Note that the inner product $\langle \mathbf{p}, \mathbf{u} \rangle = \sum_{k=1}^{d_{out}} \mathbf{p}_k \mathbf{u}_k$, it is easy the obtain the gradient $[\frac{\partial \langle \mathbf{p}, \mathbf{u} \rangle}{\partial \mathbf{F}}]_{ij} = \sum_{k=1}^{d_{out}} \frac{\partial \mathbf{p}_k}{\partial \mathbf{F}_{ij}} \mathbf{u}_k$.

To simply the current notation, we denote $\tilde{\mathbf{F}} \triangleq SF(\mathbf{F})$. According to the chain rule of derivative, we have

$$\frac{\partial \mathbf{p}_k}{\partial \mathbf{F}_{ij}} = \sum_{t=1}^{d_{out}} \frac{\partial \mathbf{p}_k}{\partial \tilde{\mathbf{F}}_{tk}} \frac{\partial \tilde{\mathbf{F}}_{tk}}{\partial \mathbf{F}_{ij}} = \sum_{t=1}^{d_{out}} \boldsymbol{\Delta}_{\mathbf{s},t} \frac{\partial \tilde{\mathbf{F}}_{tk}}{\partial \mathbf{F}_{ij}} \overset{(a)}{=} \boldsymbol{\Delta}_{\mathbf{s},i} \frac{\partial \tilde{\mathbf{F}}_{ik}}{\partial \mathbf{F}_{ij}} \overset{(b)}{=} \boldsymbol{\Delta}_{\mathbf{s},i} \tilde{\mathbf{F}}_{ik} [\delta_{kj} - \tilde{\mathbf{F}}_{ij}], \tag{12}$$

where $\delta_{kj}$ is Dirac function (equals 1 only if $k = j$, otherwise 0;), equality $(a)$ holds since softmax function is row-wise operation, and equality $(b)$ is based on Lemma C.1. Furthermore, we can obtain the gradient of fairness objective w.r.t. node presentation as follows:

$$[\frac{\partial \langle \mathbf{p}, \mathbf{u} \rangle}{\partial \mathbf{F}}]_{ij} = \sum_{k=1}^{d_{out}} \frac{\partial \mathbf{p}_k}{\partial \mathbf{F}_{ij}} \mathbf{u}_k = \sum_{k=1}^{d_{out}} \boldsymbol{\Delta}_{\mathbf{s},i} \tilde{\mathbf{F}}_{ik} [\delta_{kj} - \tilde{\mathbf{F}}_{ij}] \mathbf{u}_k = \boldsymbol{\Delta}_{\mathbf{s},i} \tilde{\mathbf{F}}_{ij} \mathbf{u}_j - \boldsymbol{\Delta}_{\mathbf{s},i} \tilde{\mathbf{F}}_{ij} \sum_{k=1}^{d_{out}} \tilde{\mathbf{F}}_{ik} \mathbf{u}_k. \tag{13}$$

Therefore, the matrix formulation is given by

$$\frac{\partial \langle \mathbf{p}, \mathbf{u} \rangle}{\partial \mathbf{F}} = \mathbf{U}_s \odot SF(\mathbf{F}) - \mathrm{Sum}_1(\mathbf{U}_s \odot SF(\mathbf{F})) SF(\mathbf{F}). \tag{14}$$

where $\mathbf{U}_s \triangleq \boldsymbol{\Delta}_s^\top \mathbf{u} \in \mathbb{R}^{n \times d_{out}}$ and $\mathrm{Sum}_1(\cdot)$ represents the summation over column dimension with preserved matrix shape. Therefore, the computation complexity for gradient $\frac{\partial \langle \mathbf{p}, \mathbf{u} \rangle}{\partial \mathbf{F}}$ is $O(nd_{out})$.

## D    Proof of Proposition 3.1

As for the proximal operators, we provide the close form in the following proposition:

**Proposition D.1** (Proximal Operators)**.** *The proximal operators $prox_{\beta h_f^*}(\mathbf{u})$ satisfies*

$$prox_{\beta h_f^*}(\mathbf{u})_j = sign(\mathbf{u})_j \min\left(|\mathbf{u}_j|, \lambda_f\right), \tag{15}$$

*where $sign(\cdot)$ and $\lambda_f$ are element-wise sign function and hyperparameter for fairness objective. In other words, such a proximal operator is an element-wise projection into $l_\infty$ ball with radius $\lambda_f$.*

We firstly show the conjugate function for general norm function $f(\mathbf{x}) = \lambda ||\mathbf{x}||$, where $\mathbf{x} \in \mathbf{R}^{1 \times d_{out}}$. The conjugate function of $f(\mathbf{x})$ satisfies

$$f^*(\mathbf{y}) = \begin{cases} 0, & ||\mathbf{x}||_* \leq \lambda, \\ +\infty, & ||\mathbf{x}||_* > \lambda. \end{cases} \tag{16}$$

where $||\mathbf{x}||_*$ is dual norm of the original norm $||\mathbf{x}||$, defined as $||\mathbf{y}||_* = \max_{||\mathbf{x}|| \leq 1} \mathbf{y}^\top \mathbf{x}$. Considering the conjugate function definition $f^*(\mathbf{y}) = \max_{\mathbf{x}} \mathbf{y}^\top \mathbf{x} - \lambda ||\mathbf{x}||$ the analysis can be divided as the following two cases:

❶ If $||\mathbf{y}||_* \leq \lambda$, according to the definition of dual norm, we have $\mathbf{y}^\top \mathbf{x} \leq ||\mathbf{x}||||\mathbf{y}||_* \leq \lambda||\mathbf{x}||$ for $\forall ||\mathbf{x}||$, where the equality holds if and only if $||\mathbf{x}|| = 0$. Hence, it is easy to obtain $f^*(\mathbf{y}) = \max_{\mathbf{x}} \mathbf{y}^\top \mathbf{x} - \lambda||\mathbf{x}|| = 0$.

❷ If $||\mathbf{y}||_* > \lambda$, note that the dual norm $||\mathbf{y}||_* = \max_{||\mathbf{x}|| \leq 1} \mathbf{y}^\top \mathbf{x} > \lambda$, there exists variables $\hat{\mathbf{x}}$ so that $||\hat{\mathbf{x}}|| \leq 1$ and $\hat{\mathbf{x}}^\top \mathbf{y} < \lambda$. Therefore, for any constant $t$, we have $f^*(\mathbf{y}) \geq \mathbf{y}^\top(t\mathbf{x}) - \lambda||t\mathbf{x}|| = t(\mathbf{y}^\top \mathbf{x} - \lambda||\mathbf{x}||) \overset{t\to\infty}{\longrightarrow} \infty$.

Based on the aforementioned two cases, it is easy to get the conjugate function for $l_1$ norm (the dual norm is $l_\infty$), i.e., the conjugate function for $h_f(\mathbf{x}) = \lambda||\mathbf{x}||_1$ is given by

$$h_f^*(\mathbf{y}) = \begin{cases} 0, & ||\mathbf{x}||_\infty \leq \lambda, \\ +\infty, & ||\mathbf{x}||_\infty > \lambda. \end{cases} \tag{17}$$

Given the conjugate function $h_f^*(\cdot)$, we further investigate the proximal operators $\mathrm{prox}_{h_f^*}$. Note that $\mathrm{prox}_{h_f^*}(\mathbf{u}) = \arg\min_{\mathbf{y}} ||\mathbf{y} - \mathbf{u}||_F^2 + h_f^*(\mathbf{y}) = \arg\min_{||\mathbf{y}||_\infty \leq \lambda_f} ||\mathbf{y} - \mathbf{u}||_F^2 = \arg\min_{\substack{\mathbf{y}_j \leq \lambda_f \\ \forall j \in [d_{out}]}} \sum_{j=1}^{d_{out}} |\mathbf{y}_j - \mathbf{u}_j|^2$, the proximal operator problem can be decomposed as element-wise sub-problem, i.e.,

$$\mathrm{prox}_{h_f^*}(\mathbf{u})_j = \arg\min_{\mathbf{y}_j \leq \lambda_f} |\mathbf{y}_j - \mathbf{u}_j|^2 = sign(\mathbf{u}_j)\min(|\mathbf{u}_j|, \lambda_f)$$

which completes the proof.

## E  More discussion on Transparency in Fairness

We aim to provide a more precise statement on transparency in fairness (TIF) and then point out why many fair methods can not achieve transparency in fairness. Intuitively, TIF represents that the influence of sensitive attribute in the inference stage for a fair method can be obtained with only a well-trained fair model and test data. Although many fair methods relying on sensitive attribute are developed to achieve a fair model, the process of how the sensitive attribute makes the model to be fair is still black-box. To this end, we introduce TIF, a general concept beyond graph data. Denote training dataset $\mathcal{D}_{train} = \{X_{train}, s_{train}, y_{train}\}$ and test dataset $\mathcal{D}_{test} = \{X_{test}, s_{test}, y_{test}\}$, where $X_{train}$ ($X_{test}$), $s_{train}$ ($s_{test}$), and $y_{train}$ ($y_{test}$) represent the input attributes, sensitive attributes, and label for model training (test). We first provide a formal statement on the influence of sensitive attribute and TIF for a specific fair method.

**What is the influence of sensitive attributes in the inference stage?**  The influence of sensitive attributes can be regarded as the difference between the well-trained fair and vanilla model. The fair model $f_{\theta^*}(\cdot)$ can be obtained using training dataset (including sensitive attribute) and a specific fair method (e.g., fair regularization, adversarial debiasing) while vanilla model $f_{\theta_0}(\cdot)$ is obtained without any usage of sensitive attribute (e.g., vanilla loss and no data pre-processing and post-processing). Define $M(f_\theta, \mathcal{D}_{test})$ as the measurement (not necessarily scaler) for a well-trained model $f_\theta(\cdot)$ given test dataset $\mathcal{D}_{test}$. For example, test loss or model prediction can be instantiations as measurements. Then the influence of sensitive attributes represents the measurement difference between the well-trained fair and vanilla models $M(f_{\theta^*}, \mathcal{D}_{test}) - M(f_{\theta_0}, \mathcal{D}_{test})$.

**What is TIF?**  TIF represents that the influence of sensitive attributes can be obtained via the well-trained fair model and test data (without access to the training data). To obtain the influence of sensitive attribute, the fair model and vanilla model are both required to obtain the influence of sensitive attribute. In other words, for existing fair methods (e.g., pre-processing, in-processing, and post-processing methods), it is intractable to obtain such influence if only having access to the fair model since the training data or vanilla model can not be accessed.

**Difference with model interpretability.**  Model interpretability aims to understand and explain the steps and decisions of the model when making predictions. There are two types of interpretability, named intrinsical interpretability and post-hoc interpretability. Intrinsically interpretable models (such as decision trees) can provide human-understandable decision-making from the model itself, while post-hoc interpretability requires external methods to help humans understand how the model makes predictions. Similar

to intrinsical interpretability, whether the fair model with TIF is essentially binary. The key difference is that TIF aims to understand how sensitive attribute helps to achieve a fair model for a specific fair method. Such a fairness-achieving process is essentially dynamic while model interpretability is static for model prediction.

**The main idea to achieve TIF.** Note that many existing methods, including pre-processing, in-processing, and post-processing methods, can not achieve TIF, we try to integrate sensitive attribute information into the forward propagation for model prediction. In this way, the influence of sensitive attributes can be obtained through model inference. Thanks to the unified optimization framework for GNNs, we develop a fair message passing (FMP), which explicitly and separately uses sensitive attributes in the (last) debiasing stage of forward propagation. which makes it easy to identify the influence. In this way, the influence of sensitive attribute can be identified using the input and output of the debiasing stage.

## F   Training Algorithms

We summarize the training algorithm for FMP and provide the pseudo codes in Algorithm 1.

---
**Algorithm 1** FMP Training Algorithm

---
**Input:** Graph dataset $=(\mathbf{X}, \mathbf{A}, \mathbf{Y})$; The total epochs $T$; Hyperparameters $\lambda_s$ and $\lambda_f$.
**Output:** The well-trained FMP model.
Initialize model parameters.
**for** epoch from 1 to $T$ **do**
    Conduct feature transformation using MLP
    Conduct propagation and debiasing as steps ❶-❺
    Calculate the cross entropy loss for node classification task
    Conduct backpropagation step to update model weight
**end for**

---

## G   Dataset Statistics

For a fair comparison with previous work, we perform the node classification task on three real-world datasets, including Pokec-n, Pokec-z, and NBA. The data statistical information on three real-world datasets is provided in Table 3. It is seen that the sensitive homophily are even higher than the label homophily coefficient among three real-world datasets, which validates that the real-world datasets is usually with large topology bias.

Table 3: Statistical Information on Datasets

| Dataset | # Nodes | # Node Features | # Edges | # Training Labels | # Training Sens |
|---------|---------|-----------------|---------|-------------------|-----------------|
| Pokec-n | 66569   | 265             | 1034094 | 4398              | 500             |
| Pokec-z | 67796   | 276             | 1235916 | 5131              | 500             |
| NBA     | 403     | 95              | 21242   | 156               | 246             |

## H   More Experimental Results

### H.1   More Experimental Setting Details

In FMP implementation, we first use 2 layers of MLP with 64 hidden units and the output dimension for MLP is 2. We also stack 2 layers for propagation and debiasing steps, where there are not any trainable model parameters. As for the model training, we adopt cross-entropy loss function with 300 epochs. We also adopt Adam optimizer with 0.001 learning rate and $1 \times 10^5$ weight decay for all models. The hyperprameters for FMP is $\lambda_f = \{0, 5, 10, 15, 20, 30, 100\}$ and $\lambda_s = \{0, 0.01, 0.1, 0.5, 1.0, 2.0, 3, 5, 10, 15, 20\}$ .

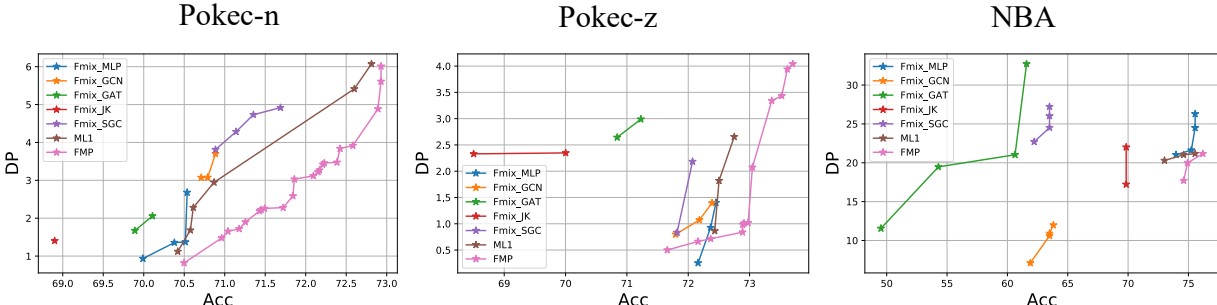

Figure 3: DP and Acc trade-off performance on three real-world datasets compared with (manifold) Fair Mixup.

## H.2 Comparison with Fair Mixup

We also implement Fair mixup (Chuang & Mroueh, 2021) as the additional baseline for different GNN backbones in Figure 3. Note that input fair mixup requires calculating model prediction for mixed input batch, it is non-trivial to adopt input fair mixup in our experiments (node classification task) since forward propagation in GNN aggregates information from neighborhoods while the neighborhood information for the mixed input batch is missing. Therefore, we adopt manifold fair mixup for the logit layer (the previous layers contain aggregation step) in our experiments. Experimental results show that our method can still achieve better accuracy-fairness tradeoff performance on three datasets.

## H.3 Sensitive Attribute Influence Probe

As for lending fairness perceptron, it represents the influence of sensitive attributes that could be identified. For example, our proposed FMP includes three steps, i.e., transformation, aggregation, and debiasing, where the sensitive attribute is explicitly adopted in debiasing step. If we aim to identify the influence of sensitive attributes for FMP, it is sufficient to check the difference between the input and output for the debiasing step. It is worth mentioning that the required information for identifying the influence of sensitive attributes is naturally from the forward propagation. Additionally, if we aim to identify the influence of sensitive attributes for existing methods (e.g, adding regularization and adversarial debiasing), the well-trained fair model is insufficient and we need additional vanilla (unfair) model without using any sensitive attribute information. In other words, these methods require model retraining with sensitive attribute movement, and thus much more resources for sensitive attributes influence auditing. The key drawback of these methods is due to encoding the sensitive attributes information into well-trained model weights. From the auditors' perspective, it is quite hard to identify the influence of sensitive attributes only given a well-trained fair model. Instead, our designed FMP explicitly adopts the sensitive attribute information in the forward propagation process, which naturally avoid the dilemma that sensitive attributes are encoded into well-trained model weight.

Figure 4 shows the visualization results for training with/without (left/right) sensitive attributes for FMP and several baselines (with GCN backbones) across three real-world datasets. From the visualization results, we observe that all methods with sensitive attribute information achieve better fairness since the logit layer representation for different sensitive attributes is mixed with each other. Therefore, it is hard to identify the sensitive attribute based on the representation and thus leads to higher fairness results. The key difference is that the results for training with/without (left/right) sensitive attribute in FMP can both be obtained through forward propagation, while the other baseline methods require model retraining to probe the influence of sensitive attributes.

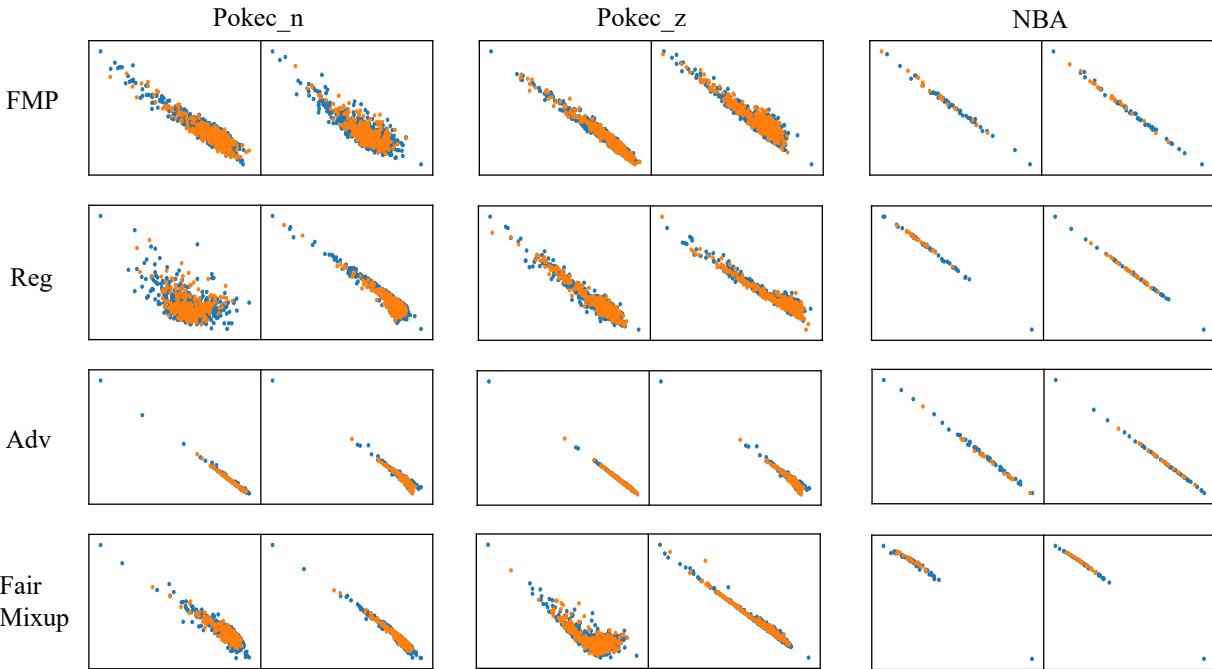

Figure 4: The visualization of logit layer node representation for training with/without (left/right) sensitive attribute for FMP and several baselines across three real-world datasets. The data point with different colors represents different sensitive attributes.

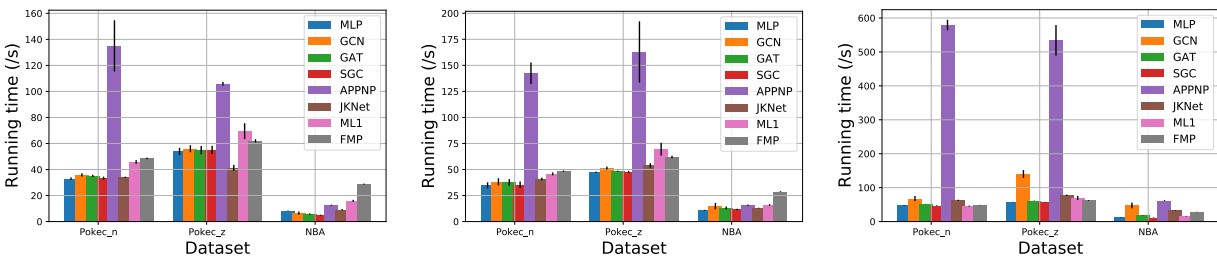

Figure 5: The running time comparison.

## H.4 Running Time Comparison

We provide running time comparison in Figure 5 for our proposed FMP and other baselines, including vanilla, regularization, and adversarial debiasing on many backbones (MLP, GCN, GAT, SGC, and APPNP). To achieve a fair comparison, we adopt the same Adam optimizer with 200 epochs with 5 running times. We list several observations as follows:

- The running time of proposed FMP is very efficient for large-scale datasets. Specifically, for the vanilla method, the running time of FMP is higher than most lighten backbone MLP with 46.97% and 15.03% time overhead on Pokec-n and Poken-z datasets, respectively. Compared with the most time-consumption APPNP, the running time of FMP is lower with 64.07% and 41.45% time overhead on Pokec-n and Poken-z datasets, respectively.

- The regularization method achieves almost the same running time compared with the vanilla method on all backbones. For example, GCN with regularization encompasses higher running time with 6.41% time overhead compared with the vanilla method. Adversarial debiasing is extremely time-consuming. For example, GCN with adversarial debiasing encompasses higher running time with 88.58% time overhead compared with the vanilla method.

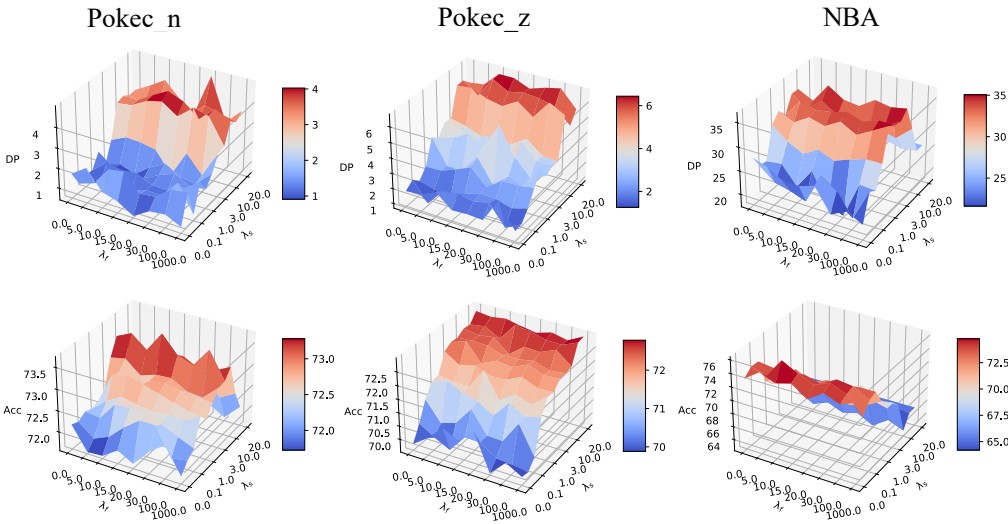

Figure 6: Hyperparameter study on fairness and smoothness hyperparameter for demographic parity and Accuracy.

## H.5 Hyperparameter Study

We provide hyperparameter study for further investigation on fairness and smoothness hyperparmeter on prediction and fairness performance on three datasets. Specifically, we tune hyperparameters as $\lambda_f = \{0.0, 5.0, 10.0, 15.0, 20.0, 30.0, 100.0, 1000.0\}$ and $\lambda_s = \{0.0, 0.1, 0.5, 1.0, 3.0, 5.0, 10.0, 15.0, 20.0\}$. From the results in Figure 6, we can make the following observations:

- The accuracy and demographic parity are extremely sensitive to the smoothness hyperparameter. It is seen that, for Pokec-n and Pokec-z datasets (NBA), a larger smoothness hyperparameter usually leads to higher (lower) accuracy with higher prediction bias. The rationale is that, only for graph data with a high label homophily coefficient, GCN-like aggregation with skip connection is beneficial. Otherwise, the neighbor's node representation with a different label will mislead the representation update.

- The appropriate fairness hyperparameter leads to better fairness and prediction performance trade-off. The reason is that fairness hyperparameter determines the perturbation vector update step size in probability space. Only appropriate step size can lead to better perturbation vector update.

## H.6 Results on Additional Datasets

We also conduct experiments on two new datasets (Recidivism and Credit), where the graph topology is constructed based on node features. In Recidivism, nodes are defendants released on bail from 1990 to 2009, where the nodes are connected based on the similarity of past criminal records and demographics. The task is to predict defendant is on bail or not, and the sensitive attribute is selected as "race". In the Credit dataset, credit card users (nodes) are connected based on the pattern similarity of their purchases and payments. The sensitive attribute is selected as "age", and the task is to predict whether a user will default on credit card payment. Figure. **??** demonstrates the tradeoff performance for different fair methods, including adding regularization, adversarial debiasing, and fair mixup. Experimental results show that our method can still achieve good accuracy-fairness tradeoff performance on three datasets. We also notice that MLP can achieve good tradeoff performance since the graph topology is manually constructed based on node attribute similarity.

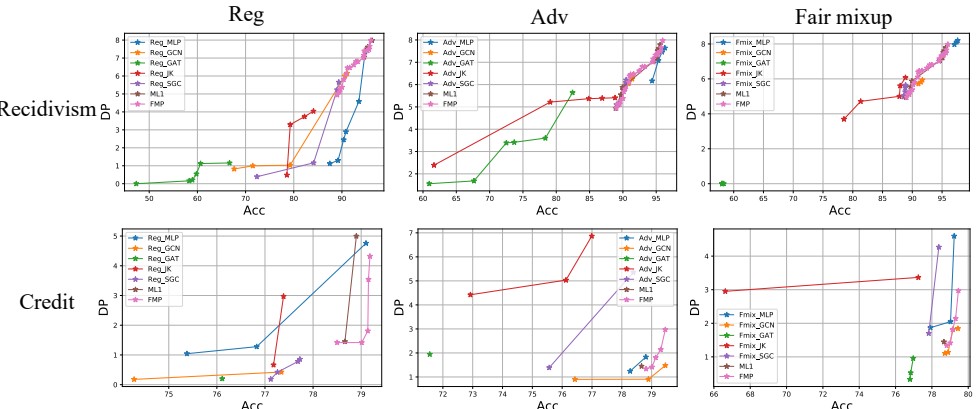

Figure 7: DP and Acc trade-off performance on three real-world datasets compared with adding regularization, adversarial debiasing, and (manifold) Fair Mixup in additional datasets.

# I Future Work

There are three lines of follow-up research directions. Firstly, achieving transparency can be further developed. For example, for the intransparent model, how can we develop external methods to probe the influence of sensitive attributes in the target model? Secondly, given the influence of sensitive attributes, how can we interpret the influence of sensitive attributes in a human-understandable way? For example, how can we measure the benefit of such influence toward fairness? Thirdly, it is also interesting to extend FMP into more general cases, such as continuous sensitive attributes (Jiang et al., 2022), and limited sensitive attributes (Dai & Wang, 2021).

# J Broader Social Impact and Limitations

Transparency in fairness is an advanced property in the fairness domain and poses huge challenges for research and industry. Many existing works mainly rely on specific fairness metrics to evaluate the prediction bias. Transparency may stimulate maintainers and auditors of machine learning systems to rethink fairness evaluation/auditing. Only achieving a fair model with a lower bias for specific fairness metrics is insufficient. The maintainers should also consider how to leverage the influence of sensitive attributes for auditors. Transparency may lead maintainers to pay more effects to improve the transparency of the fair model and could be helpful to convince the auditors. The limitations of this work are that it requests sensitive information in the inference stage.

