# OpenReview forum: "Fair Graph Message Passing"
_TMLR — Rejected by TMLR_

### Review · Reviewer_tBy5 · 2023-04-24

**Summary Of Contributions:**

This paper proposes a fair scheme for message passing in GNNs. It achieves this by leveraging the framework of graph signal denoising, and then attaching a fairness-oriented term in the associated optimisation objective. A scalable algorithm is then proposed for optimising this objective, achieving solid results on three benchmark datasets without sacrificing fairness compared to several baseline methods.

**Audience:**

Yes

**Broader Impact Concerns:**

No concerns.

**Claims And Evidence:**

Yes

**Requested Changes:**

Regarding clarity, the paper is currently somewhat difficult to read, and makes overly strong claims at times.
* The abstract has a sentence starting "Notably, FMP explicitly rendering..." which is not grammatical, and should be fixed.
* There are many typos scattered through the document -- I would recommend detailedly checking the writing in the text.
* Further, the paper is generally math-heavy, and an uninitiated reader would probably benefit from placing a descriptive figure significantly earlier in the document (e.g. page 2).
* The paper makes several strong and potentially misleading claims, especially claims that might indicate the paper "solved" the fairness problem (which is unlikely to be completely true, even given the provided results). Two example passages include: _“In this work, we achieve fairness in graphs from the model architecture perspective”_, and _"We demonstrate proof-of-concept that a meticulously crafted GNN architecture can achieve fairness for graph data"_. Both of these claims are very strong, and imply that fairness has been fully achieved. I highly recommend such passages be toned down to avoid confusion and misrepresentation.
* Minor: There is a "Figure ??" in the Appendix.

Regarding the method's generality, it appears to rests on the graph signal denoising perspective. From my understanding, this perspective can explain GNN methods that are "convolutional" in nature, in the sense that they directly aggregate neighbour-dependent messages (SGC, GCN, GAT, APPNP). However, it is unclear to me whether such a framework would work for the fully-general message-passing GNNs (such as MPNNs (Gilmer et al.) and Graph Networks (Battaglia et al.)). Given that the method's name is _'fair message passing'_, it is in my opinion quite critical to demonstrate that methods like MPNN can also be made fair in this way.

Going beyond this context, it appears that the method presented as 'FMP' in the experiments corresponds to a specific propagation rule (GCN?). Would different propagation rules yield different FMP updates? If so, they should all be ablated in the paper. If not, the paper should make this clear.

Lastly, I believe that a paper oriented on fairness would strongly benefit from comprehensive qualitative results, demonstrating to the reader specific cases in which unfair predictions are avoided. Perhaps the authors can showcase predictions on some specific subgraphs of the datasets they study, where a baseline method (such as GCN or GAT) would unfairly classify the nodes, and FMP would behave fairly. Perhaps with some explanation on how FMP avoided unfair behaviour in these specific examples? Such studies would likely be quite valuable for future work.

**Strengths And Weaknesses:**

The proposed method is sound, and addresses an important and timely problem. I believe the claims are well-supported by experimental evidence. In principle, I believe it should be accepted for TMLR.

My main concerns with the paper are the lack of clarity, potentially unclear generality of the method, and lack of strong qualitative analysis. Please see the "Requested Changes" section for more details, and a summary of suggested changes for the authors.

---

### Review · Reviewer_Ahy9 · 2023-04-26

**Summary Of Contributions:**

This manuscript introduces a novel graph neural network (GNN) architecture, denoted Fair Message Passing (FMP), that is aimed at promoting fairness. This architecture relies on the message passing paradigm to aggregate information and explicitly incorporates the use of sensitive attributes to ensure the expression center of population demographic group nodes. The empirical findings from the experiments conducted on three real-world datasets demonstrate that the proposed FMP architecture outperforms other baselines in both fairness and accuracy. Additionally, the authors present a discussion on the extension to fairness loss. This research contributes to the advancement of fairness in GNNs by providing a fresh perspective.




**Audience:**

Yes

**Broader Impact Concerns:**

NA.

**Claims And Evidence:**

Yes

**Requested Changes:**

Please see the weakness section to improve the experiments.

**Strengths And Weaknesses:**

``Pros:``
- The paper is generally well-written.
- Code is provided.

I'm generally happy with this submission, with only a few minor concerns.

``Cons:``
- Experiments are not that sufficient. It would be great if the results on some general datasets like OGB datasets can be provided to further demonstrate the effectiveness of the proposed FMP.
- Please consider comparing with more state-of-the-art GNN baselines in the experiments.

``Minor:``
- Typos: “Notably, FMP explicitly *rendering* (renders) sensitive attribute …”
- Please remove the dot in the title of Sec. 3.2.1.

---

### Review · Reviewer_VmER · 2023-04-27

**Summary Of Contributions:**

The paper proposes a new graph neural network (GNN) architecture called Fair Message Passing (FMP) that aims to achieve fairness in graphs by mitigating bias in node representations. FMP consists of two steps: aggregation and bias mitigation. In the aggregation step, FMP uses a standard GNN layer to update node features by combining information from their neighbors. In the bias mitigation step, FMP pushes the representations of nodes belonging to different demographic groups closer together, using a regularization term based on the distance between group centroids. The paper shows that FMP can achieve a better trade-off between fairness and accuracy than several baselines on three real-world datasets for node classification tasks. The paper also provides a theoretical analysis of FMP from the perspectives of model interpretation, efficiency, and white-box usage of sensitive attributes.

**Audience:**

Yes

**Broader Impact Concerns:**

The authors discussed transparency in their broader impact statement, which could be misleading since it is not explicitly defined or discussed in the manuscript under fair graph learning settings. Is the mentioned transparency equal to the white-box usage that the authors discussed in Section 4?

**Claims And Evidence:**

Yes

**Requested Changes:**

Please kindly refer to Section Strengths and Weaknesses for detailed suggestions.

**Strengths And Weaknesses:**

Strengths:
* It demonstrates that a carefully designed GNN architecture can achieve fairness for graph data, without relying on data pre-processing or fair training strategies, which are common approaches in the literature.
* It introduces a simple and effective way of mitigating bias in node representations, which could also be accelerated by exploiting the softmax property during optimization.
* It provides empirical evidence that FMP can improve both fairness and accuracy on various datasets and tasks, compared to existing methods.

Weaknesses:
* The writing can be further improved. For example, $F$ start to appear in Section 2 but is defined later in Section 3 after Eq. (1), which could be confusing for readers. Similarly, $\alpha$ is not defined from paragraph PPNP/APPNP. There is also an unresolved reference on Page 21 (Figure ??). The authors are suggested to further proofread to make the manuscript self-contained.

* The motivation is relatively vague from the Section Introduction. While it is interesting to explore how the model architecture contributes to fairness graph learning, why the model architecture is better than the other two aspects is not explicitly discussed. Given no baselines from existing works based on graph pre-processing or fair training strategies for regular graphs (note that the selected regularization and adversarial debiasing methods in experiments are not designed for regular graphs) are compared during experiments, it is hard to empirically find out the superior of sole model architecture-based methods.

* One possible advantage of the proposed method is to bring white-box usage for sensitive attributes, which is an ability that is not shared by both graph pre-processing or fair training strategies. However, this advantage is not straightforward to understand. Could the authors provide an example with practical scenarios to demonstrate how to employ this white-box usage during practice?

* In addition to this white-box usage, the authors mention that if we aim to identify the influence of sensitive attributes for FMP, it is sufficient to check the difference between the input and output for the debiasing step. The authors are suggested to include empirical ablation studies to support this claim.

---

### Review · Reviewer_cDbm · 2023-05-02

**Summary Of Contributions:**

The paper presents a new graph neural network (GNN) architecture called Fair Message Passing (FMP). It aims to achieve fairness in graphs by mitigating bias in node representations. The paper is well presented and the source code is made available. See details in below.

**Audience:**

Yes

**Claims And Evidence:**

No

**Requested Changes:**

Questions on Technical details:
1.$\mathbf{F}$ is not clearly defined and confusing. I found the definition in [2] that $\mathbf$ is a clean signal which is need to be recovered. The author should write the preliminary part more clearly following [2] since the notation of graph signal processing sometimes is different from GNNs’.
2.The optimization of model in Sec3.2 is still confusing. I think the author can write the overall objective function in a closed form.
3.What is the difference between step 2 in FMP and the addition of a normalization term? In the step 2, moving the model one step in the direction of reducing <p, u> and adding a regularization term concerning <p, u> can achieve a similar effect.
4.How to handle discrete computations in step 4 using backward propagation algorithm?
5.How to make sure the debiasing procedure in step 4 will not influence the accuracy? From the Figure 1, the proximity of orange and blue nodes inevitably makes them more challenging to distinguish.
6.In summary of contributions, the author says, “We propose FMP to achieve fairness via explicitly incorporating sensitive attribute information in message passing”. However, the method solely considers labels of demographic parity, neglecting sensitive feature information such as age. The statement can be revised.

**Strengths And Weaknesses:**

Pros:
1.The study of fairness-aware message passing provides good insights for graph learning community.
2.The paper is well-structured and easy to follow.
3.The code is available.

Cons:
Writing:
1.The literature review is notably incomplete. Contrary to the author's claim that the GNNs architecture perspective for improving fairness in graphs is less explored, numerous works have proposed GNNs ensuring fairness. For details, please refer to the relevant survey [1].
2.Additionally, the author described GAT, GCN, and GraphSAGE from the perspective of graph signal processing in the Preliminary section. This aspect has been elaborated in detail by [2] and is not highly relevant to the main focus of this paper, fairness.
3.There exists grammar mistakes and typos.

Experiments is not convincing:
1.The experimental settings and datasets are consistent with previous studies on the fairness of GNNs. However, no comparison with any fair GNN is made in the experimental section. For example, when compared to FairGAT [3], it is slightly inferior on both Poke-z and Pokec-n datasets. Specifically, on the NBA dataset, FairGAT’s $\Delta_{eo}$ is only 0.7, while FMP reaches 13.33. Can the author explain the reason for this large difference? Comprehensive comparison with recent methods should be conducted.
2.The author compares with Adversarial Debiasing and Regularization, but the referenced methods do not directly study GNN fairness. The authors should compare with fair GNNs that utilize Adversarial Training, such as [3], which employs adversarial training to filter sensitive information and subsequently enhance fairness. [4] introduce regularization of topological bias in learning objective.
3.As Equation 1 consists of fairness and smoothness objectives, it would be beneficial to conduct an ablation study to investigate their roles.

[1] A Survey on Fairness for Machine Learning on Graphs.
[2] A unified view on graph neural networks as graph signal denoising.
[3] Improving Fairness in Graph Neural Networks via Mitigating Sensitive Attribute Leakage.
[4] TAM: Topology-Aware Margin Loss for Class-Imbalanced Node Classification

---

### Decision · Action_Editors · 2023-07-19

**Recommendation:** Reject

**Comment:**

This paper introduces the FMP model architecture to address fairness in GNN. While it presents decent mathematical derivation and empirical study, improvements are necessary to make it acceptable. The motivation from the GNN architecture perspective lacks elaboration, and the novelty against prior research from this perspective is overclaimed. Additional experiments with more datasets and relevant baselines, along with ablation studies, are needed to demonstrate the superiority of FMP. The writing can also be enhanced by correcting typos, clarifying notations and details, simplifying preliminaries, and enriching the literature review.

As the authors haven't responded or revised the manuscript, the AE recommends rejection.

**Audience:**

Researchers working on graph neural networks or AI fairness would have interests in this paper.


**Claims And Evidence:**

As most of the reviewers commented (also summarized in the comment below), while the main experimental results support a fairness gain of the proposed method, there still remain several claims in this submission that are inaccurate or not well supported.